# Synthesis and Self-Assembly of Conjugated Block Copolymers

**DOI:** 10.3390/polym13010110

**Published:** 2020-12-29

**Authors:** Lin-Lin Xiao, Xu Zhou, Kan Yue, Zi-Hao Guo

**Affiliations:** 1School of Molecular Science and Engineering, South China Advanced Institute for Soft Matter Science and Technology, South China University of Technology, Guangzhou 510640, China; 201920155214@mail.scut.edu.cn (L.-L.X.); zhouxupku@126.com (X.Z.); kanyue@scut.edu.cn (K.Y.); 2Guangdong Provincial Key Laboratory of Functional and Intelligent Hybrid Materials and Devices, South China University of Technology, Guangzhou 510640, China

**Keywords:** conjugated block copolymers, block copolymers, self-assembly, microphase separation

## Abstract

In the past two decades, conjugated polymers (CPs) have drawn great attention due to their excellent conductivity and charge mobility, rendering them broad applications in organic electronics. Controlling over the morphologies and nanostructures of CPs is very important to improve the performance of CP-based devices, which is still a tremendously difficult task. Conjugated block copolymers (cBCPs), composed of different CP blocks or CP coupled with coiled polymeric blocks, not only maintain the advantages of high conductivity and mobility but also demonstrate features of morphological versatility and tunability. Due to the strong π–π interaction and crystallinity of the conjugated backbones, the self-assembly behaviors of cBCPs are very complicated and largely remain to be explored. In this tutorial review, we first summarize the general synthetic methods for different types of cBCPs. Then, recent studies on the self-assembly behaviors of cBCPs are discussed, with an emphasis on the structural factors that affect the morphologies of cBCPs both in bulk and thin film states. Finally, we briefly provide our outlook on the future research of the self-assembly of cBCPs.

## 1. Introduction

Conjugated polymers (CPs) have alternating single and double bonds in the polymer backbone, making the polymer chain rigid and conductive. In 2000, the Noble Prize in Chemistry was awarded to MacDiarmid, Heeger, and Shirakawa in recognition of their discovery of highly conductive polyacetylene [1]. Despite relatively low charge mobility and conductivity compared to inorganic materials, conjugated polymers have extraordinary advantages of solution processability, low cost, light weight, structural versatility, and easy functionality. CP-based organic electronic materials experienced rapid development during the past twenty years and have found broad applications in the field of organic light-emitting diodes (OLEDs) [2,3,4,5], organic field-effect transistors (OFETs) [6,7,8], organic photovoltaics (OPVs) [9,10], and organic thermoelectrics (OTEs) [11,12], etc. 

It has been widely accepted that the performance of CP-based devices is greatly influenced by the micro/nano morphologies in bulk or in thin films. For example, many CPs, e.g., poly(3-hexylthiophene) (P3HT) [13], show better charge mobility when polymer chains pack in the “edge-on” orientation (Figure 1a). For OPV devices, the ideal morphology is the interpenetrated lamellar structure, which is beneficial to the charge separation and transportation of electrons and holes to different electrodes (Figure 1b) [14]. As a result, great efforts have been devoted to investigate the controlled formation of desired morphologies. Many factors, such as annealing condition, polymer solution concentration, selection of solvent, mole ratio of binary system, molecular weight, etc., can affect the morphologies of CPs in devices. Therefore, optimizing the morphologies of the active CP-containing layer is critical and one of the most time-consuming steps in device fabrication. In this context, conjugated block copolymers (cBCPs) have been proposed as an ideal solution to address this issue, since cBCPs might combine the advantages of high conductivity and mobility of CPs and morphological versatility and tunability of block copolymers (BCPs).

BCPs are comprised of two or more polymeric blocks with different chemical compositions connected by covalent bonds. One of the most remarkable and fascinating features of BCPs is that they can self-assemble into different ordered micro/nano structures both in bulk and in solutions, which makes them good candidates for practical or potential applications at various areas, such as biomaterials, biomedicine, porous materials, nanolithography, organic electronics, etc. [15,16,17,18,19,20,21,22,23,24]. The self-assembly of BCPs in bulk has been extensively studied since 1960s, and the experimental and theoretical knowledge is well established [25,26,27,28,29,30,31,32,33].

In the simplest case of diblock polymers, as shown in Figure 2a, due to the immiscibility of two different blocks, they can self-assemble into different ordered morphologies including spheres (S), cylinders (C), bicontinuous gyroids (G), lamellae (L), etc. The phase separation behaviors of BCPs depend on three parameters: (1) *f*, the volume fractions of the A and B blocks (*f*_A_ and *f*_B_, with *f*_A_ + *f*_B_ = 1); (2) *N*, total degree of polymerization (*N*_A_ + *N*_B_ = *N*); (3) *χ*, the Flory–Huggins interaction parameter between two polymer blocks. The driving force of the phase separation is the Gibbs energy change in the self-assembly process, which can divide into mixing enthalpy and mixing entropy. Unlike binary mixture of small molecules or polymer solutions, the mixing entropy of BCPs is usually very small. Thus, the product of *N* and the *χ* parameter, which describes the degree of thermodynamic incompatibility between the two chemically different polymeric blocks, governs the phase separation behaviors in BCP self-assembly. Several theories have been developed to predict the phase separation behaviors of BCP in bulk [25,34]. Nowadays, the self-consistent mean-field (SCMF) theory can predict the phase diagram of BCPs in good agreement with experimental data, as shown in Figure 2b,c. For example, symmetric diblock copolymers (*f*_A_ = *f*_B_ = 0.5) self-assemble into lamellar morphology when the product *χN* value is larger than the order–disorder transition values ((*χN*)_ODT_ = 10.5) [26]. In addition, many nonclassical spherical packing phases, e.g., Frank–Kasper phases, have also been predicted by SCMF [35] and later experimentally discovered in recent years [36]. 

Compared to traditional BCPs, the study on cBCPs is still in the seminal stage, despite many recent progresses. A flexible polymer chain is usually considered as a Gaussian chain with a coil configuration, while the polymer chain of CPs is rigid and has a rod-like configuration. Distinguished from the conventional “coil–coil” type BCPs, cBCPs can be divided into two categories, namely, the “rod–coil” and “rod–rod” types (Figure 3). The rigid rod-like backbone of CPs leads to very different self-assembly behavior of cBCPs that are distinct from conventional “coil–coil” BCPs. For example, the (*χN*)_ODT_ for symmetric “rod–coil” and “rod–rod” BCPs is 8.5 and 8.2, respectively, as predicted by R. Borsali et al. [37]. This result indicates that cBCPs containing a rigid block have stronger trend to phase separate than the “coil–coil” BCPs. In addition, conjugated polymer backbones have a strong tendency to form crystalline structures, and the melting temperature (*T*_m_) of crystalline blocks may also affect the morphology and size of the phase separation. When the block segregation strength is very high ((*χ*N)/(*χN*)_ODT_ > 3), crystallization is restricted in the region of phase separation; when the block segregation strength is lower than the critical inter-block segregation strength ((*χ*N)/(*χN*)_ODT_ < 1.5), break-through crystallization occurs to induce the phase separation; when the segregation strength is in an intermediate range ((*χ*N)/(*χN*)_ODT_ = 1.5~3), the crystallization is not completely restricted to the area of phase separation [38]. However, the self-assembly of cBCPs is very complicated in a real situation due to the strong π–π interaction between conjugated polymer backbones.

In this review, we focus on synthesis of cBCPs, their self-assembly behaviors, and influencing factors for their micro/nano morphology. In the first part, we summarize the synthetic methods for cBCPs including “rod–coil” and “rod–rod”. Second, we discuss how cBCPs self-assemble into different morphologies in bulk, solution and thin film. Thirdly, some influencing factors at the molecular level for cBCP self-assembly will be discussed, including side chain engineering, regioregularity of the backbone, amphiphilic, etc. We also discuss the relationship between the device performance and polymer morphologies in third part. Finally, we will give a summary and perspective for cBCPs.

## 2. Synthesis of Conjugated Block Copolymers

As mentioned above, cBCPs are divided into two categories, “rod–coil” (Figure 3b) and “rod–rod” (Figure 3c), based on the type of polymer blocks. “Rod–coil” cBCPs are composed of a rigid CP segment and a flexible polymer chain. Benefitting from the fast development of living polymerization techniques, the synthesis of “rod–coil” cBCPs are extensively studied, and people are able to synthesize various “rod–coil” cBCPs with different chemical components. Figure 4 shows representative chemical structures for rod and coil polymers. However, CPs are synthesized by metal catalyzed coupling reactions, which usually do not have the “living” feature. In this context, it is still very challenging to synthesize “rod–rod” cBCPs with well-controlled molecular weight and distribution. In this part, we will mainly focus on the synthesis of “rod–coil” and “rod–rod” cBCPs, and a few examples with different topological architecture will be also discussed.

### 2.1. Synthesis of “Rod–Coil” Conjugated Block Copolymers

Usually, two strategies are employed to synthesize “rod–coil” cBCPs: (i) polymerization from macroinitiator (the grafting-from approach) (Figure 5a), and (ii) the coupling of a pre-synthesized rod and coil block (the grafting-onto approach) (Figure 6a). The rod CP blocks are generally prepared by a transition-metal catalyzed coupling reaction, such as Kumada, Suzuki, and Stille reaction. In order to endow sufficient solubility to CPs, soluble side chains are usually introduced to the backbone, which is critical to subsequent synthesis and processing steps. The coil blocks are normally synthesized from vinyl-based monomers via living/controlled polymerization, such as ionic polymerization, atom transfer radical polymerization (ATRP) [39], reversible addition−fragmentation chain-transfer (RAFT) polymerization [40], and nitroxide-mediated free radical polymerization (NMP) [41]. 

#### 2.1.1. Grafting-From Approach

Living/controlled polymerization has been used to prepare rod–coil cBCPs from conjugated macroinitiator via the grafting-from approach. Among them, the ATRP technique is widely used due to its simplicity and applicability of various monomers. Generally, rod blocks are synthesized by transition-metal catalyzed coupling reaction and then functionalized with an ATRP initiator in the chain end for the next living/controlled polymerization step. For example, poly[2,7-(9,9-dihexylfluorene)]-*b*-poly(*tert*-butyl acrylate) (PF-*b*-P*t*BA) was synthesized from polyfluorene macroinitiator (PF-Br) by ATRP. Amphiphilic rod-coil cBCPs poly[2,7-(9,9-dihexylfluorene)]-*b*-poly(acrylic acid) (PF-*b*-P*t*AA) were obtained after excess trifluoroacetic acid (TFA) to hydrolyze PF-*b*-P*t*BA [42]. Liu et al. reported that hydroxy-terminated poly(3-hexylthiophene) (P3HT) was modified by reaction with 2-bromopropionyl bromide to generate macroinitiators, and subsequently, di- or triblock copolymers were synthesized by ATRP [43]. In addition, donor-acceptor type rod-coil cBCP, a diblock copolymer (P3HT-*b*-C_60_) based on P3HT and fullerene, was also obtained by performing ATPR on monomers that attach fullerenes to hydroxyethyl methacrylate through esterification reactions [44]. Different rod–coil cBCPs have been prepared by the ATRP method, including poly(paraphenylene vinylene)-*b*-poly(styrene) (PPV-*b*-PS) [45], poly(1,4-(2,5-dioctyloxy)phenylene vinylene)-*b*-poly(methyl methacrylate) (PPV-b-PMMA) [46], and poly[2,7-(9,9-dihexylfluorene)]-*b*-poly[2-(dimethylamino)ethyl methacrylate] (PF-*b*-PDMAEMA) [47].

Other living/controlled polymerizations, such as living ionic polymerization, RAFT polymerization, ring-opening polymerization, and NMP, can be also employed to synthesize “rod–coil” cBCPs after installing their corresponding initiators at the end of the rod block. For example, a series of P3HT-based cBCPs were synthesized by using a trithiocarbonate-terminated P3HT macromolecular RAFT agent [48]. Stalmach et al. reported that poly(2,5-dioctyloxy-1,4-phenylene vinylene)-*b*-poly(*n*-butylacrylate) (PPV-*b*-P*n*BA) was prepared from an acyclic nitroxide PPV macroinitiator [49]. Dai and coworkers used vinyl end-functionalized P3HT combined with sec-butyllithium to perform the anionic polymerization of 2-vinyl pyridine monomer and finally obtained poly(3-hexylthiophene)-*b*-poly(2-vinyl pyridine) (P3HT-*b*-P2VP) [50]. In addition, Park et al. synthesized poly(3-octylthiophene)-*b*-poly (ethylene oxide) (POT-*b*-PEO), which uses hydroxyl-terminated POT and then initiates anionic ring-opening polymerization of ethylene oxide under the catalysis of potassium naphthalide [51].

Ring-opening polymerization (ROP) and ring-opening metathesis polymerization (ROMP) have been also used to prepare “rod–coil” cBCPs. For example, hydroxyl end-functionalized poly(3-alkylthiophene)s (P3ATs) were used as macroinitiators for the controlled ROP of d,l-lactide to afford poly(3-alkylthiophene)-*b*-polylactide (P3AT-*b*-PLA) [52]. Radano et al. reported that block copolymers poly(3-hexylthiophene)-*b*-polyethylene (P3HT-*b*-PE) were synthesized through the chain transfer of olefin-terminated P3HT in the presence of cyclooctene via ring-opening metathesis polymerization (ROMP) [53].

Similarly, “coil” block-based macroinitiators are prepared by chain-end functionalization and subsequently used for synthesizing “coil–rod” cBCPs. For example, Cuendias et al. also report a synthetic route of using a coil block as an initiator to polymerize the rod part. They synthesized a bromine-containing thiophene derivative to perform ATRP on the end of the bromine and then conducted oxidative polymerization on the thiophene part to afford poly(3-hexylthiophene)-*b*-poly(n-butyl acrylate) (P3HT-*b*-P*n*BA) [54]. However, this strategy is not widely applied due to non-trivial synthesis. 

#### 2.1.2. Grafting-Onto Approach

For the grafting-onto approach, a “coil” block and “rod” block are synthesized individually by different types of polymerization. Then, the as-prepared two blocks are functionalized with a specific group in the chain end, which can be coupled with each other (Figure 6a). A highly efficient “click” reaction is usually used for the grafting-onto approach. Polymers synthesized by ATRP are easily introduced to the azide group in the chain end by nucleophilic substitution reaction and then react with alkyne functionalized CPs by the copper (I) catalyzed “click” reaction to afford “rod–coil” cBCPs. For example, Li et al. synthesized poly(3-hexylthiophene)-*b*-poly(*tert*-butyl acrylate) (P3HT-*b*-PtBA) by ethynyl end functionalized P3HT and azide functionalized PtBA, in which PtBA was prepared by ATRP [55]. Lohwasser et al. modified the alkynyl group at the end of P3HT, which was followed by a click reaction with poly(4-vinylpyridine)(P4VP) polymerized by NMRP to obtain P3HT-*b*-P4VP [56]. Saejin et al. also used P3HT modified with an alkynyl group to react with azide functionalized PEG to obtain P3HT-*b*-PEG through a click reaction [57]. Other reactions are also employed for the grafting-onto approach, such as esterification, nucleophilic addition, condensation reaction, etc. For example, oligo(phenylene vinylene)-*b*-poly(ethylene glycol)(OPV-*b*-PEO) could be prepared through the esterification reaction between OPV with a carboxyl end and poly(ethylene glycol) monomethyl ether [58]. Olsen et al. reported the synthesis of poly(2,5-di(2′-ethylhexyloxy)-1,4-phenylene vinylene)-*b*-poly(1,4-isoprene) (PPV-*b*-PI). Benzene solution of aldehyde-terminated PPV was injected to quench an anionic polymerization of isoprene to afford the desired PPV-*b*-PI [59].

### 2.2. Synthesis of “Rod–Rod” Conjugated Block Copolymers

Compared to “rod–coil” cBCPs, the synthesis of “rod–rod” cBCPs is still very challenging, and many reported studies in the literature mainly focus on the poly(3-alkylthiophene) (P3AT) system prepared by Grignard metathesis (GRIM) polymerization (Figure 7a) [60,61,62,63,64,65,66]. GRIM polymerization, also known as Kumada catalyst transfer polymerization (KCTP), was developed by McCullough’s group in 1999 [67]. Due to its living nature, GRIM polymerization provides a versatile tool to synthesize regioregular poly(3-alkylthiophene) (P3AT) with well-controlled molecular weight, narrow distribution, and good chain-end functionality [68]. For example, Lee and coworkers used sequential GIRM polymerization to obtain poly(3-dodecylthiophene)-*b*-poly(3-(2-(2-(2-methoxyethoxy)ethoxy)ethoxy)methyl thiophene) copolymer (P3DDT-*b*-P3TEGT) [65]. GRIM polymerization can also be used to synthesize donor–acceptor type “rod–rod” cBCPs. Block copolymers poly(3-hexylthiophene)-*b*-poly(benzotriazole) (P3HT-*b*-PBTz) were prepared by the sequential monomer addition of benzotriazole and thiophene [66].

Other types of transition metal catalyzed coupling reactions have also been used to prepared “rod–rod” cBCPs (Figure 7b) [69,70,71,72,73,74]. Lin et al. employed GRIM and Suzuki–Miyaura polycondensation to prepared a seriels of “donor–acceptor” cBCPs including P3HT-*b*-poly(9′,9′-dioctyl fluorene) (P3HT-*b*-PF), P3HT-*b*-poly(9′,9′-dioctyl fluorene -*alt*- benzothiadiazole) (P3HT-*b*-PFBT), and P3HT-*b*-poly(2,7-(9′,9′-dioctyl-fluorene)-*alt*-5,5-(4′,7′-di-2-thienyl-2′,1′,3′-benzothiadiazole) (P3HT-*b*-PFTBT) [71]. Ku et al. also used bromine-terminated P3HT and AA + BB monomer to conduct Stille polymerization to prepare poly(3-hexylthiophene)-*b*-poly(diketopyrrolopyrrole-terthiophene) (P3HT-*b*-DPP) [72].

In recent years, a new type of “rod–rod” cBCPs was reported. Unlike fully conjugated “rod–rod” block copolymers, two conjugated blocks are connected by a flexible short alky chain to afford donor–σ–acceptor type “rod–rod” cBCPs. For example, Lee et al. prepared two types of cBCPs, PTQI-*b*-PNDIS and PTQI-*b*-PNDISL, containing quinoxaline-thiophene (PTQI) and naphthalene dicarboximide-selenophene (PNDIS) blocks, in which PTQI-*b*-PNDIS is fully conjugated and PTQI-*b*-PNDISL is composed of a donor and acceptor block connected by non-conjugated alkylene spacer [75].

### 2.3. Other Types of cBCPs

cBCPs with complex architectures have been reported, such as conjugated multi-block copolymers [76,77,78], miktoarm star copolymers [79,80,81,82,83], polymer brushes [84,85], dendrimers [86], cyclic polymers [87], etc. Similar to binary cBCPs, polymers of other architectures can also be synthesized by using the grafting-from approach and grafting-onto approach. For the grafting-from approach, Lee et al. reported PDMSB-*b*-PS-*b*-PMMA synthesized by sequential anionic polymerization [78]. There are also some miktoarm star copolymers prepared by combining grafting-from and grafting-onto, such as (P3HT)_2_ PMMA [82], (P3DDT)_2_PMMA [80], PMMA_2_P3HT [81], etc. For polymer brushes, it is usually used to modify the polymer on norbornene to form a macromonomer and then synthesize conjugated polymer brushes through ROMP, such as P3HT-*b*-PLA brush copolymers [85]. With the current development of synthesis methods, there are relatively few studies on the synthesis and phase separation of complex architectures, and this field still needs to be expanded.

## 3. Self-Assembly of cBCPs

### 3.1. In Bulk

A lot of research has been published on the topic of “rod–coil” BCP self-assembly. “Rod–coil” BCPs show remarkably rich structural morphologies in bulk, such as lamellar phases, spherical, cylindrical, smectic, C-type smectic, O-type structure (HPL), hexagonal puck phases, hexagonally perforated layer structures, etc. [50,88,89,90,91]. Theoretical calculations also predict that A15 and gyroid phases could also exist in “rod–coil” BCPs. In the study of PPV-*b*-P4VP, Sary et al. found that they can form spherical, hexagonal columnar, lamellar, and smectic phases, based on volume fraction of PPV [92]. These results are consistent with theoretical predictions (Figure 8) [93]. However, for the conjugated “rod” blocks, they have a strong π–π interaction and are easy to crystallize and thus form fiber structures. The self-assembly of cBCPs is more complicated and different from “rod-coil” BCPs. The interaction between the anisotropic conjugated rods should be considered.

For “rod–rod” cBCPs, only a few examples were reported. Due to the strong interaction “rod–rod” between blocks, i.e., π–π interaction and crystallization, it is difficult to prepare ordered nanostructures. The morphologies of “rod–rod” cBCPs are affected by the competition between rod–rod interaction and phase separation. The theory of “rod–rod” cBCP self-assembly is not yet clear, and their phase separation is generally independent of its specific chemical structure and composition. Noting that due to the high rigidity and poor curl ability of rod blocks, aggregates with curvature, e.g., spherical, are seldom formed in “rod–rod” cBCPs [94].

### 3.2. In Solution

One key research area of cBCPs is their self-assembly behaviors in solution. Amphiphilic “rod–coil” cBCPs will self-assemble into various nanostructures depending on the molecular structures. Usually, amphiphiles with a hydrophobic chain and a hydrophilic head could self-assemble into spherical micelle, cylindrical micelle, vesicle, and lamellae (Figure 9), which can be predicted by the packing parameter, *P = v/(a*_0_*l_c_*), in which *v* is the volume of the hydrophobic chain, *a*_0_ is the polar head surface at the critical micelle concentration (cmc), and *l_c_* is the hydrophobic chain length [95]. Based on this theory, the aggregates of amphiphilic “rod–coil” cBCPs in solution could be tuned by changing the ratio of rod/coil. Kim et al. reported that a series of carbohydrate conjugate rod−coil amphiphiles with different rod/coil ratios are able to self-assemble into spheres, vesicles, and cylinders [96,97]. Selective solvent concentration is another parameter that affects the morphologies of amphiphilic “rod–coil” cBCPs in solution. For example, poly(phenylquinoline)-*b*-polystyrene(PPQ-*b*-PS) formed lamellar structures in dichloromethane/trifluoroacetic acid(DCM/TFA) (*v/v* = 1) mixed solvent and cylinders in DCM/TFA (*v/v* = 1/9) at room temperature. TFA is a good solvent for the conjugated PPQ block and protonates its imine nitrogen, while the coil PS block is insoluble in TFA [98]. Tung et al. reported that PF-*b*-PAA formed a tape-like structure when the coil part is short. When PAA block increased, various morphologies were observed with methanol concentration [42]. In addition to rod/coil ratios and selective solvent concentration, there are also other parameters that drive the self-assembly of “rod–coil” cBCPs in solution including pH values, temperature, the presence of additives and block compositions, etc. Several comprehensive reviews related to “rod–coil” cBCPs have been published [99,100,101,102,103,104].

### 3.3. In Thin Film

Semiconducting CPs can be processed in organic solvents, and their electronic properties are largely determined by the nanostructure of the film. The orientation of polymer backbone is very important and has a remarkable effect on the charge mobility. Sirringhaus et al. found that mobility of edge-on orientated P3HT was two orders of magnitude of face-on orientated P3HT in 1999 [105]. Due to the anisotropy of transport properties, when semiconducting CPs self-assemble into edge-on lamellar morphology, charge transportation is more favorable compared to face-on orientation. Serials of “rod–coil” cBPCs were synthesized and their morphologies were investigated, including P3HT-*b*-PS, P3HT-*b*-PI, P3HT-*b*-PMMA, P3HT-*b*-PEG, etc. For example, Oh et al. designed a P3HT-*b*-PEG polymer in which PEG was amphiphilic and could cause self-assembly at a gas–liquid interface. P3HT-*b*-PEG thin films prepared on an inclined water surface have a long-range order and direction-controlled P3HT nanowire arrays, which have better hole mobility than P3HT-*b*-PEG thin films prepared on a flat water surface [57]. Han et al. found that P3DDT-*b*-PLA could self-assemble into lamellar morphology in thin film after solvo-microwave annealing, in which a P3DDT block has an edge-on orientation [106].

In “rod–coil” cBCPs, the coil block are generally insulating polymers and therefore will lower the carrier concentration, limiting their device performance. In this context, all conjugated “rod–rod” cBCPs have become a focus of interest. Several conjugated “rod–rod” cBCPs have been synthesized with excellent charge-transporting mobility, and their thin film morphologies were also investigated [61,62,63,107,108,109,110]. Zhang et al. synthesized a series of highly regioregular poly(3-hexylthiophene-*b*-3-(2-ethylhexyl)thiophene)s (P3HT-*b*-P3EHT) and found that the P3EHT block could induce the self-assembly of P3HT in thin film (Figure 10a) [62]. For poly(*p*-phenylene)-*b*-(3-hexylthiophene) (PPP-*b*-P3HT), Yang et al. discovered that edge-on and face-on orientation could be tuned by different annealing processes (Figure 10c) [111]. End-on orientation is also achieved by “rod–rod” cBCPs. Lee et al. prepared poly(3-dodecylthiophene)-*b*-poly(3-(2-(2-(2-methoxyethoxy)ethoxy)ethoxy)methyl thiophene) copolymer (P3DDT-*b*-P3TEGT). A P3DDT-*b*-P3TEGT thin film showed parallel oriented lamellar microdomains to the substrate in which the polymer backbones are perpendicular to the substrate and thus show improved hole mobility along the vertical direction compared with P3DDT (Figure 10b) [65].

For binary donor–acceptor mixed bulk heterojunction (BHJ) solar cells, it is very challenging to fully control the nano-phase separation in order to achieve better exciton dissociation at the donor–acceptor interface. Using cBCP is considered to be an ideal solution to resolve this problem, even though great efforts have been made to tune the morphology of the active layer, including the choice of casting solvent, solvent evaporation rate, and thermal or solvent annealing. In 2000, Stalmach et al. reported a diblock copolymer composed of PPV and polystyrene functionalized with fullerenes(PSFu), in which PS blocks are partially functionalized with fullerene. In a spin-cast film of PPV-*b*-PSFu, a micrometer-scale, honeycomb-like nanostructure is obtained, and luminescence from PPV is quenched indicating efficient electron transfer to C_60_ [112]. Zhang et al. report that “rod–coil” donor–acceptor copolymer poly(thiophene-*b*-perylene diimide) copolymer P3HT-*b*-PDI formed fibrillar morphology in thin film after solvent vapor annealing and gave a power conversion efficiency of 0.49% in a solar cell device. A donor–acceptor type “rod–rod” block can self-assemble into a donor–acceptor alternating lamellar nanostructure, which is beneficial for exciton dissociation in organic solar cells. Guo et al. designed and prepared a novel donor–acceptor type poly(3-hexylthiophene)-*b*-poly((9,9-dioctylfluorene)-2,7-diyl-alt-[4,7-bis(thiophen-5-yl)-2,1,3-benzothiadiazole]-2′,2″-diyl) (P3HT-*b*-PFTBT) block copolymer. By using P3HT-*b*-PFTBT as an acting layer, 3% efficiency was achieved for OPV devices. X-ray scattering results demonstrated that the high device performance of P3HT-*b*-PFTBT was due to self-assembly into lamellar structures with primarily face-on crystallite orientations [14].

## 4. Factors that Affect the Self-Assembly of cBCPs

Many factors will affect the self-assembly of cBCP. Similar to conventional “coil–coil” BPCs, *f*, *N,* and *χ* could be adjusted to tune the morphology of cBCPs. In addition, other parameters, such as side chains, regioregularity of the conjugated backbone, block–block connecting linkers, etc., will also affect cBCP self-assembly. 

### 4.1. Volume Fraction (f)

Similar to “coil–coil” BCPs, the morphologies of cBCPs can be tuned by adjusting the volume fraction of each block [50,113]. For example, Dai et al. reported that a series of well-defined P3HT-*b*-P2VP, synthesized by GRIM and anionic polymerization, showed sphere, cylinder, lamellae with an increase of P3HT content and finally, a nanofiber structure appeared when the mass fraction of P3HT was greater than 50% [50]. Sary et al. found that the morphologies of PPV-*b*-P4VP changed from a lamellar structure to spherical structure as the *f*_P4VP_ increased (Figure 11) [92]. Lee et al. also reported that P3HT-*b*-P2VP formed a lamellar structure when *f*_P2VP_ = 0.30–0.60 and the copolymer exhibited cylindrical and spherical phases when *f*_P2VP_ = 0.60–0.80. The phase diagram of P3HT-*b*-P2VP was constructed based on the experimental results, noting that the gyroid phase of cBCP was first found in P3HT-*b*-P2VP (*f*_P2VP_ = 0.68) [113].

### 4.2. Block–Block Interactions (χ and μ)

For cBCPs, the morphology is not only governed by enthalpic interaction, which is defined as a Flory–Huggins interaction parameter (*χ*), but it is also affected by the Maier–Saupe parameter (*μ*), which characterizes the rod–rod interactions. For example, Lee et al. prepared a series of P3HT-*b*-P2VP block copolymers [113]. When *χN*_total_ > *μN*_P3HT_, the polymer tends to microphase separation to produce various microphase structures including lamellae, cylinder, sphere, and gyroid; otherwise, the polymer forms a nanofiber morphology. Especially, for donor–acceptor cBCPs, the donor and acceptor block have a strong interaction, which makes *χ* very small or even negative, leading to a difficulty of self-assembly.

### 4.3. Degree of Polymerization (N)

The microphase separation from the traditional coil–coil copolymer suggests that increasing N promotes phase separation. Kynaston et al. synthesized Poly(3-dodecylthiophene)-*b*-poly(3-dodecylselenophene) (P3DDT-*b*-P3DDS) with a block ratio of approximately 1:1 (Figure 12). When the degree of polymerization is relatively low (*N* < 35), lamellar structures are formed due to the crystallization of each block. However, when N is increased, thiophene and selenophene blocks co-crystallize into fibers (*N* = 50–60) and patchy fiber (*N* > 80). Field-effect transistors testing showed that co-crystalline fibers have the best device performance [114].

### 4.4. Side Chain Engineering

In the synthesis of CPs, soluble side chains are usually introduced to the rigid polymer backbone not only because they can endow polymers with sufficient solubility but also they can tune the packing and morphologies of polymer chains [115,116,117]. P3HT has a strong π–π interaction and tends to crystallize at room temperature. Replacing hexyl with a longer or branched side chain will largely decrease π–π interaction, leading to micro phase separation. Lin et al. synthesized three different poly(3-alkylthiophene)-*b*-poly(methyl methacrylate)(P3AT-*b*-PMMA)s by changing the length of the side chain of thiophen. At approximately *f*_P3AT_ = 0.5, P3AT-*b*-PMMA showed versatile morphology depend on the side chain length of thiophene, and the *χ* value was also calculated (Figure 13a) [118]. Guo et al. found that all conjugated block copolymers, P3HT-*b*-PFTBT, self-assemble into a small domain size (<10 nm) after introducing a side chain into the PFTBT unit, leading to poor OPV performance. This probably is because the miscibility of P3HT and PFTBT has become better, resulting in weaker phase separation [14].

All conjugated donor–acceptor BCPs always suffer from poor ability to phase separate due to the strong interaction between the donor and acceptor blocks. Introducing immiscible side chains into different blocks, which can increase the *χ* between the donor and acceptor blocks, can resolve this problem. For example, Mitchell et al. designed an amphiphilic block copolymer P3HT-*b*-PFTBT with a hydrophilic tetraethylene glycol side chain attached to the acceptor [35]. This structure can better control the microphase separation and can form well-defined structures in thin film. However, the performance of devices prepared from this material was unexpectedly low, which was probably due to poor electron transport through steric-induced twisting of the acceptor backbone. Adding fluorine-containing side chains can also increase the incompatibility between blocks. Lombeck et al. prepared SF-PCDTBT-*b*-P3HT in which the side chain on the donor unit was fluorinated (Figure 13b). This material has a strong tendency of phase separation due to the introduction of semi-fluorinated side chains. A bi-continuous structure, which is beneficial for charge separation and charge transfer, is formed, leading a power conversion efficiency of 1% when using this block polymer as an active layer in a solar cell device [119].

### 4.5. Regioregularity

For P3AT-based “rod–coil” cBCPs, the morphologies in bulk or thin film can be modulated by controlling over the regioregularity (RR) of the P3AT block. Kim et al. designed a series of P3HT-*b*-P2VP with different RR. High RR P3HT-*b*-P2VP has a very strong interchain interaction and is preferential to crystalline, which makes it difficult to form an ordered nanostructure. In contrast, crystallization is inhibited in low RR P3HT-*b*-P2VP and well-ordered nanostructures are achieved, i.e., lamellar and cylindrical phases [120]. They also found that the domain space of poly(3-dodecylthiophene)-*b*-poly(2-vinylpyridine) (P3DDT-*b*-P2VP) can be tuned by regulating the RR of the P3DDT block (Figure 14) [121].

In addition, RR also affects the crystalline temperature of PAHT and subsequently has an influence on the self-assembly behaviors. For example, the crystal growth of low RR (70–80%) P3DDT-*b*-P2VP, which has a low crystallization temperature below the glass transition of P2VP, is confined by the cylindrical or lamellar BCP structure. The phase separation is dominated by rod–coil segregation. Thus, high RR (94%) P3DDT-*b*-P2VP crystalline at a temperature higher than the glass transition of P2VP and only a lamellar structure is formed, which is attributed to the geometric compatibility between crystal growth and self-assembled symmetry [122].

### 4.6. The Influence of Middle Bridge

For “rod–rod” cBCPs, the block–block connecting linker can be tuned to control the interface between the donor and acceptor domain in bulk or thin film [100]. Lee et al. synthesized two types of cBCPs (Figure 15a), which were PTQI-*b*-PNDIS with two blocks connected by a π conjugated unit and PTQI-*b*-PNDISL with two blocks connected by a non-conjugated spacer. They found that PTQI-*b*-PNDISL shows a more prominent nanophase behavior compared to PTQI-*b*-PNDIS. In addition, because of the flexible spacer buffer between the two blocks, each block appears as an independent crystallization behavior leading to the power conversion efficiency of PTQI-*b*-PNDISL as high as 1.54%, while PTQI-*b*-PNDIS is only 0.36% (Figure 15b) [75].

### 4.7. Active Nanoparticles

Adding active nanoparticles is another strategy to adjust the π–π interaction between rod blocks. Loudy et al. reported that the thin film morphologies of poly(3-hexylthiophene)-*b*-poly(ethylene glycol methyl ether methacrylate) (P3HT-*b*-PEGMA) are changed from fibrils structure into out-of-plane cylinders after incorporating gold nanoparticles with PEG ligands. This is because the crystallization of the rod block was destroyed, and the self-assembly is driven by nanophase segregation [123]. Ye et al. found that titanium dioxide (TiO_2_) nanoparticles (NPs) are preferentially confined in the P2VP domains when P3HT-*b*-P2VP mixed with nicotinic acid-modified TiO_2_ NPs. At the addition of TiO_2_ NPs to 40 wt % or higher, the morphologies of lamellar and cylindrical structures are disturbed. P3HT-*b*-P2VP/TiO_2_ hybrid materials exhibit a > 30 fold improvement in power conversion efficiency compared to the corresponding P3HT/P2VP/TiO_2_ polymer blend hybrid in a solar cell device [124].

## 5. Summary and Outlook

cBCPs have been considered as a novel type of materials with unique chemical structures, excellent electronic properties, and, more importantly, diverse self-assembly behaviors. In this review, we have summarized the recent progresses on the synthesis methods of different types of cBCPs and their self-assembly in solution, solid films, and thin films. We have also discussed the influences of several parameters on the morphology of cBCPs and the resulting electronic properties and device performance. Despite all the recent studies on cBCPs, this field still remains largely unexplored with many problems to be solved. From our perspective, first, a new synthetic methodology needs to be developed, especially for the synthesis of fully conjugated block copolymers. Second, compared to the “rod–coil” cBCPs, the “rod–rod” cBCPs have fully conjugated backbones and therefore are expected to have better conductivity and better electronic properties. However, feasible control over the morphology of such “rod–rod” cBCPs is still a grand challenge due to the strong inter-chain interactions and the stiffness of the conjugated backbone. Side-chain engineering is recognized as an effective solution to this problem. Thirdly, it is expected that the collaboration of theoretical studies with experimental investigation could provide more insights to guide rational designs to further improve the morphological control and performance of cBCP materials.

## Figures and Tables

**Figure 1 polymers-13-00110-f001:**
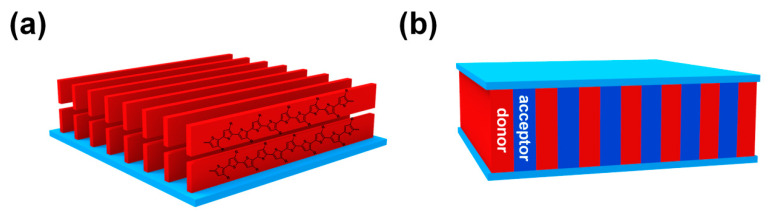
(**a**) The “edge-on” orientation of poly(3-hexylthiophene) (P3HT) that promotes charge transportation; (**b**) the proposed ideal interpenetrated lamellar structure of donor and acceptor materials in organic photovoltaics (OPV) devices.

**Figure 2 polymers-13-00110-f002:**
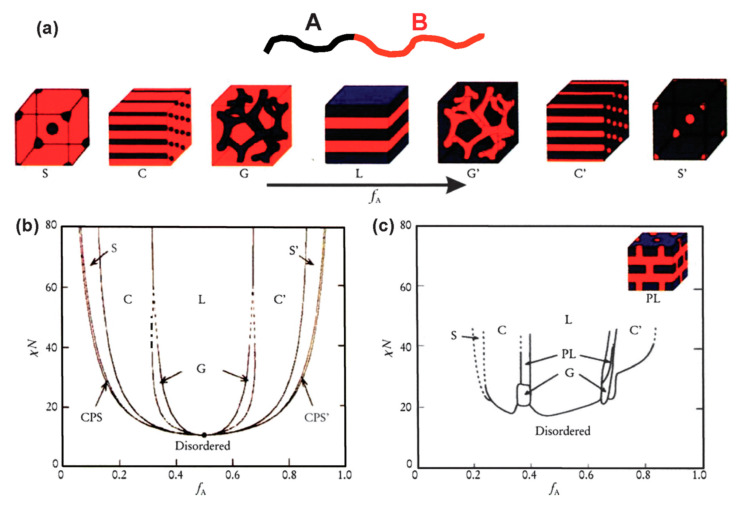
(**a**) Equilibrium morphologies of AB diblock copolymers in bulk: S and S’ = body-centered cubic spheres, C and C’ = hexagonally packed cylinders, G and G’ = bicontinuous gyroids, and L = lamellae. (**b**) Theoretical phase diagram of AB diblocks predicted by the self-consistent mean-field theory, depending on volume fraction (*f*) of the blocks and the segregation parameter, *χN*; CPS and CPS’ = closely packed spheres. (**c**) Experimental phase diagram of polystyrene-*b*-polyisoprene copolymers, in which *f*_A_ represents the volume fraction of polyisoprene, PL = perforated lamellae. (Reproduced with permission from [28], Copyright Royal Society of Chemistry).

**Figure 3 polymers-13-00110-f003:**
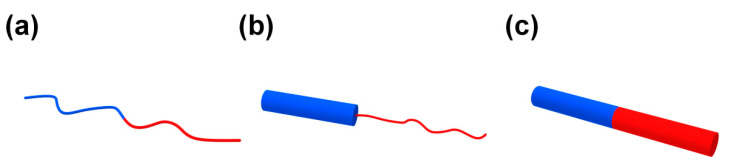
(**a**) Conventional “coil–coil” block copolymers (BCPs); (**b**) “rod–coil” conjugated block copolymers (cBCPs); and (**c**) “rod–rod” cBCPs.

**Figure 4 polymers-13-00110-f004:**
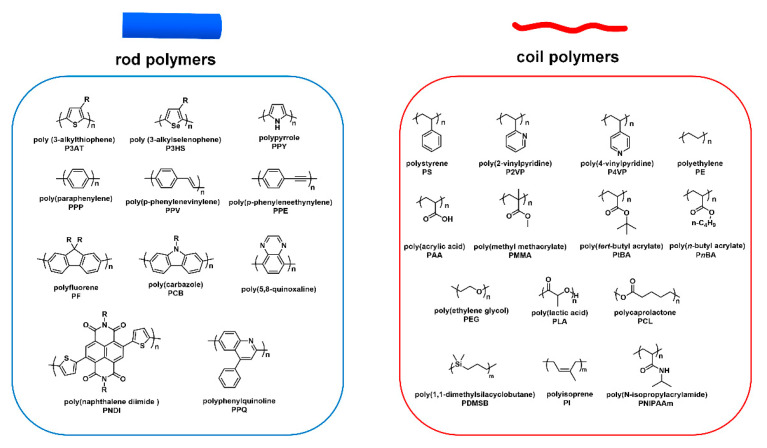
Representative chemical structures for rod and coil polymers.

**Figure 5 polymers-13-00110-f005:**
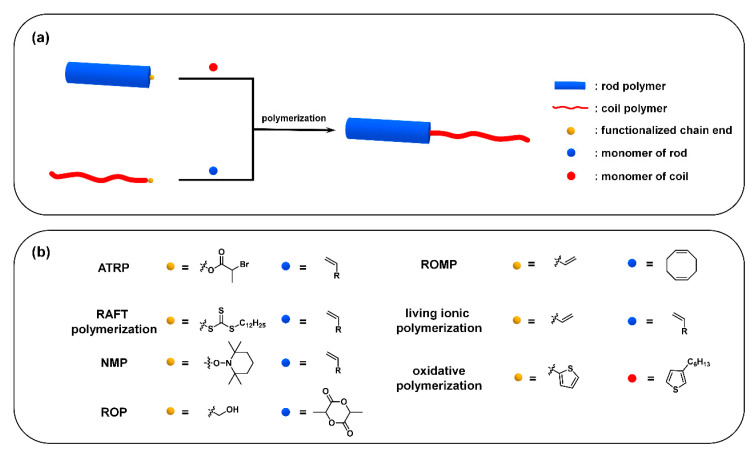
(**a**) The grafting-from approach from rod polymer and roil polymer; (**b**) representative functionalized chain end and monomer for different polymerizations.

**Figure 6 polymers-13-00110-f006:**
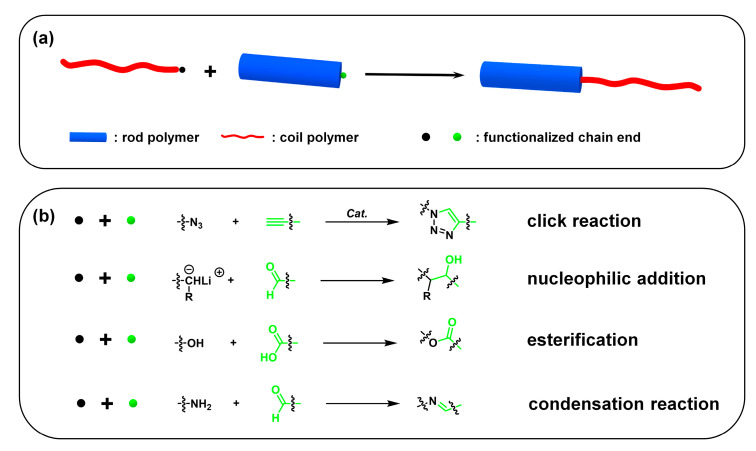
(**a**) The grafting-onto approach; (**b**) representative functionalized chain ends and reaction types.

**Figure 7 polymers-13-00110-f007:**
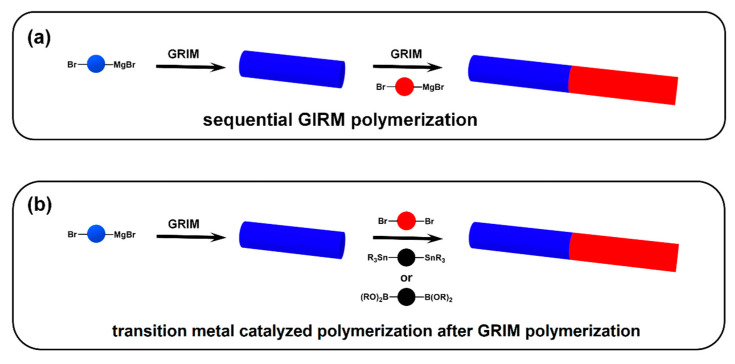
(**a**) The grafting-onto approach; (**b**) representative functionalized chain ends and reaction types.

**Figure 8 polymers-13-00110-f008:**
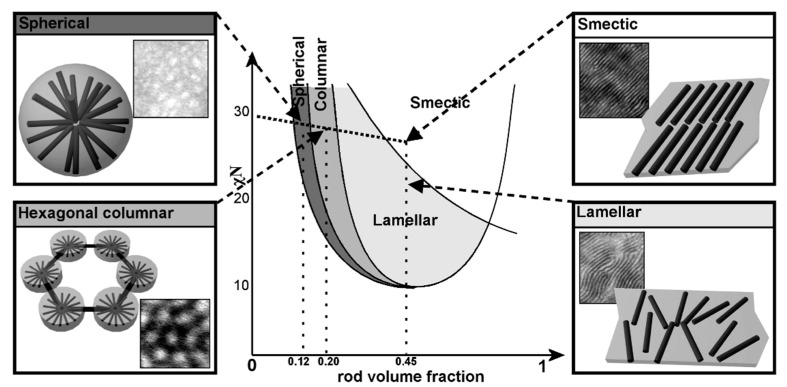
Comparison between the phase diagram predicted by Landau expansion theories for rod−coil block copolymers and the microphase-separated morphologies experimentally observed for PPV−P4VP block copolymers. (Reprinted with permission from Ref [92]. Copyright American Chemical Society).

**Figure 9 polymers-13-00110-f009:**
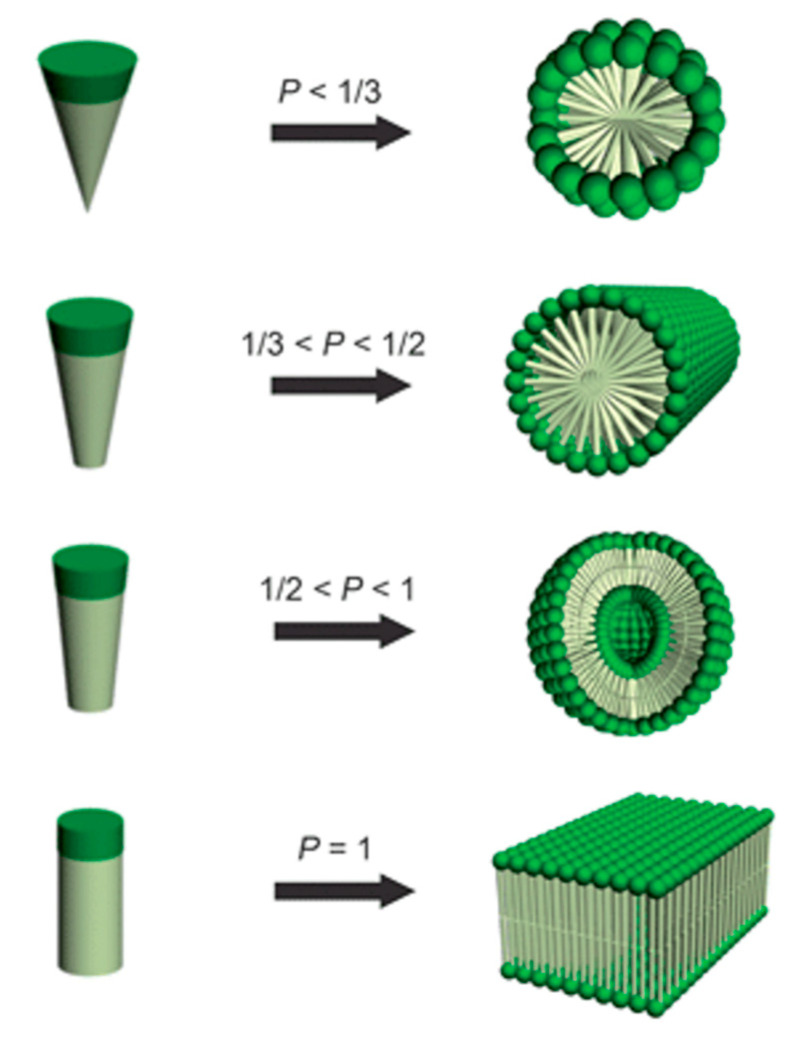
Dependence of nanostructure morphologies on the relative volume fraction of hydrophobic and hydrophilic blocks. (Reprinted with permission from Ref [95]. Copyright Royal Society of Chemistry).

**Figure 10 polymers-13-00110-f010:**
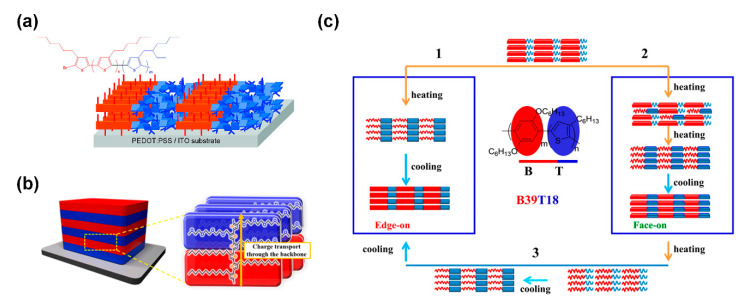
(**a**) “Edge-on” orientation of poly(3-hexylthiophene-*b*-3-(2-ethylhexyl)thiophene)s (P3HT-*b*-P3EHT) in thin film; (Reprinted with permission from Ref [62]. Copyright American Chemical Society); (**b**) “end-on” orientation of poly(3-dodecylthiophene)-*b*-poly(3-(2-(2-(2-methoxyethoxy)ethoxy)ethoxy)methyl thiophene) copolymer (P3DDT-*b*-P3TEGT) in thin film; (Reprinted with permission from Ref [65]. Copyright American Chemical Society); (**c**) three different self-epitaxial crystallization circles by controlling the heating process for poly(*p*-phenylene)-*b*-(3-hexylthiophene) (PPP-*b*-P3HT). (Reprinted with permission from Ref [111]. Copyright American Chemical Society).

**Figure 11 polymers-13-00110-f011:**
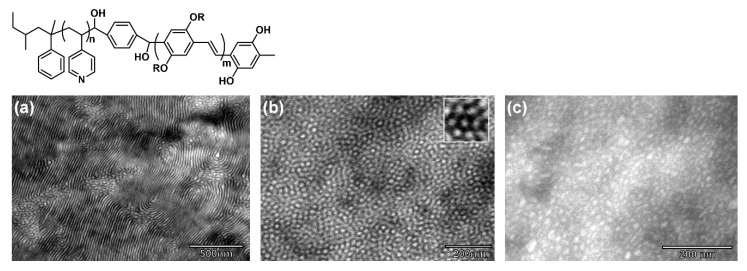
Iodine-stained TEM micrographs of (**a**) PPV_11_-*b*-P4VP_51_ (ordered lamellar structure); (**b**) PPV_11_-*b*-P4VP_173_ (ordered hexagonal phase); (**c**) PPV_11_-*b*-P4VP_295_ (poor order of spherical phase). (Reprinted with permission from Ref. [92]. Copyright American Chemical Society).

**Figure 12 polymers-13-00110-f012:**
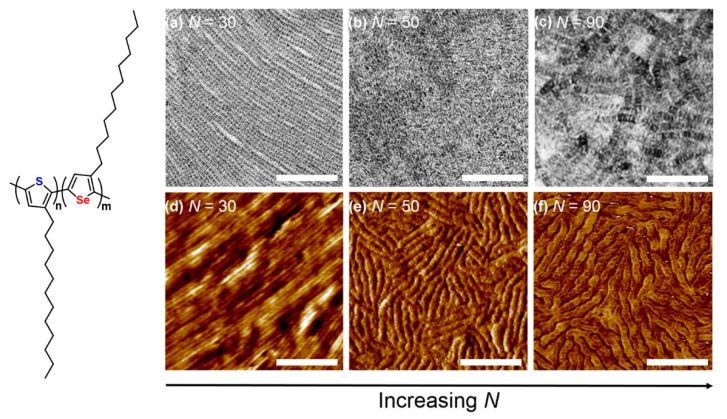
TEM micrographs and tapping-mode AFM phase images of P3DDT-*b*-P3DDS BCP-annealed thin films depicting lamellar structures ((**a**,**d**), *N* = 30), co-crystallized fibers ((**b**,**e**), *N* = 50), and a patchy fiber morphology ((**c**,**f**), *N* = 90). Scale bars are 200 nm. (Reprinted with permission from Ref. [114]. Copyright American Chemical Society).

**Figure 13 polymers-13-00110-f013:**
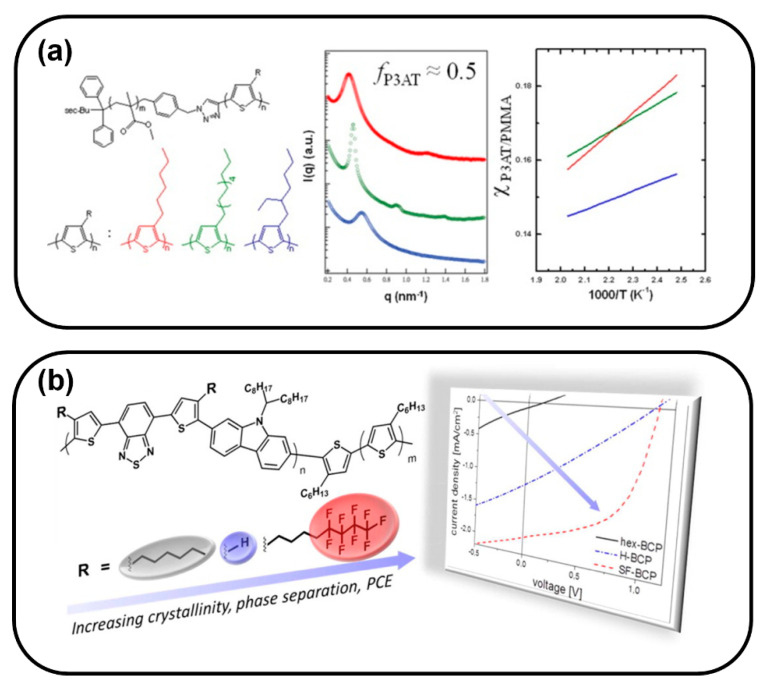
(**a**) Chemical structures of P3AT-*b*-PMMAs with different side chain length, Small-angle X-ray scattering results and temperature dependence of the Flory–Huggins interaction parameter between P3AT and PMMA segments for P3AT-*b*-PMMA; (Reprinted with permission from Ref. [118]. Copyright American Chemical Society); (**b**) chemical structure of PCDTBT-*b*-P3HT with different side chains and *J*–*V* characteristics. (Reprinted with permission from Ref. [119]. Copyright American Chemical Society).

**Figure 14 polymers-13-00110-f014:**
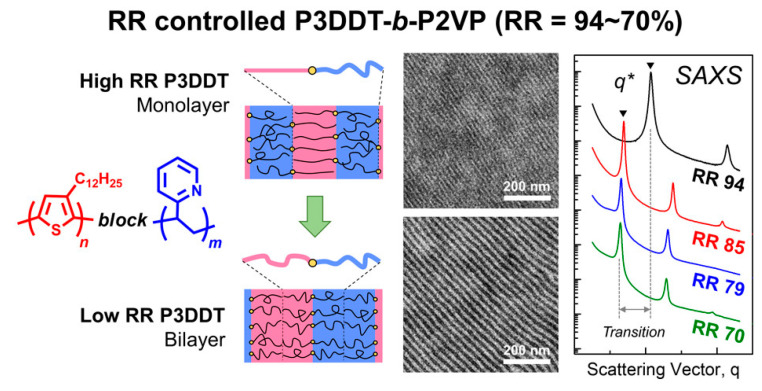
Chemical structure of P3DDT-*b*-P2VP and their morphological results from TEM and SAXS characterization. (Reprinted with permission from Ref. [121]. Copyright American Chemical Society).

**Figure 15 polymers-13-00110-f015:**
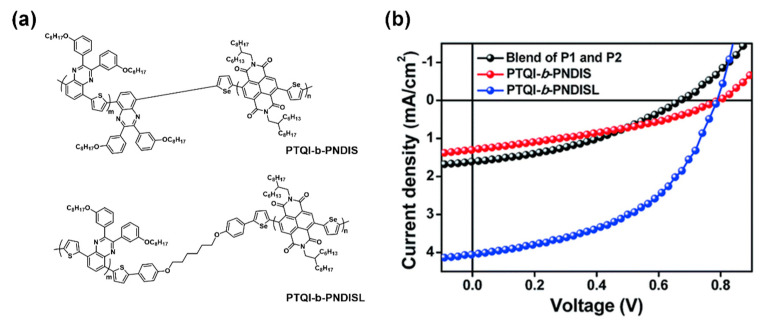
(**a**) Chemical structures of conjugated PTQI-*b*-PNDIS and donor–σ–acceptor type PTQI-*b*-PNDISL; (**b**) *J*–*V* characteristics of (red) PTQI-*b*-PNDIS, and (blue) PTQI-*b*-PNDISL. (Reprinted with permission from Ref. [75]. Copyright Royal Society of Chemistry).

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
