# Peer review of "Synthesis and Self-Assembly of Conjugated Block Copolymers"

_polymers, 2020, doi:10.3390/polym13010110_

Round 1

Reviewer 1 Report

Dear Authors,

The Authors discuss the types of conjugated block copolymers, their synthesis and self-assembly properties, along with some factors influencing this self-assembly. I believe that the subject matter is interesting, even though it has been extensively reviewed in the past (see my comment below). The Authors’ outlook on the matters presented in the manuscript is justified and reasonable.

In my opinion, apart from the issue of over-coverage of the subject, the manuscript contains only a few small issues, which should be improved, but are not of particular significance.

            As such, I recommend for the publication of the manuscript in MDPI Polymers, only pending several corrections to the manuscript, with my main recommendations and comments being listed below:

References:

            Out of the 116 works cited in the manuscript, at least one of the Authors is present in only 1 of the citations, with the presence of self-cited item being justified. In line with the Authors’ indication of their focus on the last two decades, the cited works strike a good balance, with 44 works representing 2001-2010 and 54 works representing 2011-2020 and indicating the gradual development and increasing interest in the subject over the recent years.

            However, the issue of self-assembly in block copolymers has been the subject of nigh-countless literature review articles, with more than twenty such articles being published in 2020 alone. To exemplify (DOI numbers):

  • 10.1016/j.nantod.2020.100936
  • 10.1002/marc.201900485
  • 10.1039/C9QM00444K
  • 10.1080/14686996.2018.1553108
  • 10.3390/polym12102432
  • 10.1016/j.cis.2020.102213
  • 10.1039/D0CS00021C
  • 10.1039/D0PY00455C
  • 10.1080/17425247.2020.1767582
  • 10.1016/j.progpolymsci.2020.101278

            Consequently, even though the Authors have wisely chosen to limit their review manuscript only to conjugated copolymers, I believe that this subject may still be too general to compete with other recently published works, particularly with those that focus on more specialised applications. Perhaps the scope of the review should be narrowed down even further?

Minor remarks:

Section 4:

            The subsection headings are erroneously labelled (repeat subsection 4.1, subsection 4.3 being the seventh one in the section), i.e.:

4.1 Volume Fraction (f)

4.1 Block-Block Interactions (χ and μ)

4.1 Degree of polymerization (N)

4.1 Side Chain Engineering

4.1 Regioregularity

4.1 The influence of middle bridge

4.3 Active Nanoparticles

The English used in the manuscript, apart from a number of minor mistakes (some examples are listed below), is generally satisfactory.

Line 36: “For example, many CPs, i.g.” - should be “For example, many CPs, e.g.”

Figure 4: “poly (3-alkyl selenophene)”- should be “poly(3-alkylselenophene)”

Author Response

Reviewer #1:

The Authors discuss the types of conjugated block copolymers, their synthesis and self-assembly properties, along with some factors influencing this self-assembly. I believe that the subject matter is interesting, even though it has been extensively reviewed in the past (see my comment below). The Authors’outlook on the matters presented in the manuscript is justified and reasonable.

In my opinion, apart from the issue of over-coverage of the subject, the manuscript contains only a few small issues, which should be improved, but are not of particular significance.

As such, I recommend for the publication of the manuscript in MDPI Polymers, only pending several corrections to the manuscript, with my main recommendations and comments being listed below:

Q1: Out of the 116 works cited in the manuscript, at least one of the Authors is present in only 1 of the citations, with the presence of self-cited item being justified. In line with the Authors indication of their focus on the last two decades, the cited works strike a good balance, with 44 works representing 2001-2010 and 54 works representing 2011-2020 and indicating the gradual development and increasing interest in the subject over the recent years.

However, the issue of self-assembly in block copolymers has been the subject of nigh-countless literature review articles, with more than twenty such articles being published in 2020 alone. To exemplify (DOI numbers):

  • 10.1016/j.nantod.2020.100936
  • 10.1002/marc.201900485
  • 10.1039/C9QM00444K
  • 10.1080/14686996.2018.1553108
  • 10.3390/polym12102432
  • 10.1016/j.cis.2020.102213
  • 10.1039/D0CS00021C
  • 10.1039/D0PY00455C
  • 10.1080/17425247.2020.1767582
  • 10.1016/j.progpolymsci.2020.101278

Consequently, even though the Authors have wisely chosen to limit their review manuscript only to conjugated copolymers, I believe that this subject may still be too general to compete with other recently published works, particularly with those that focus on more specialised applications. Perhaps the scope of the review should be narrowed down even further?

Our response: We apologize that some important papers were not cited in the original manuscript. The above references are included in our revised manuscript. See ref. 20, 21, 22, 23, 24, 29, 30, 31, 32, 33.

Q2:  The subsection headings are erroneously labelled (repeat subsection 4.1, subsection 4.3 being the seventh one in the section), i.e.:

4.1 Volume Fraction (f)

4.1 Block-Block Interactions (χ and μ)

4.1 Degree of polymerization (N)

4.1 Side Chain Engineering

4.1 Regioregularity

4.1 The influence of middle bridge

4.3 Active Nanoparticles

The English used in the manuscript, apart from a number of minor mistakes (some examples are listed below), is generally satisfactory.

Line 36: For example, many CPs, i.g. - should be For example, many CPs, e.g.

Figure 4: poly (3-alkyl selenophene)- should be poly(3-alkylselenophene)

Our response: Thank you for your advice. We carefully checked section number, the phrasing of the article and made corrections as suggested.

Reviewer 2 Report

This manuscript describes synthesis and self-assembly of cBCPs. In contrast, the controlling over the morphologies and nanostructures of CPs is very important to improve the performance of CP-based devices. This is a "delicious" paper I have no hesitation in recommending.

Author Response

Reviewer #2:

This manuscript describes synthesis and self-assembly of cBCPs. In contrast, the controlling over the morphologies and nanostructures of CPs is very important to improve the performance of CP-based devices. This is a "delicious" paper I have no hesitation in recommending.

Our response: We are very grateful for the reviewer’s high evaluation.